# Finding a Direct Method for a Dynamic Process: The DD (Direct and Dynamic) Cell-Tox Method

**DOI:** 10.3390/ijms25105133

**Published:** 2024-05-09

**Authors:** Eneko Madorran, Lidija Kocbek Šaherl, Mateja Rakuša, Iztok Takač, Miha Munda

**Affiliations:** 1Faculty of Medicine, Institute of Anatomy, Histology and Embryology, University of Maribor, Taborska Ulica 8, 2000 Maribor, Slovenia; lidija.kocbek-saherl@um.si (L.K.Š.); mateja.rakusa@um.si (M.R.); miha.munda@um.si (M.M.); 2Division for Gynecology and Perinatology, University Medical Centre Maribor, Ljubljanska Ulica 5, 2000 Maribor, Slovenia; iztok.takac@ukc-mb.si; 3Faculty of Medicine, University of Maribor, Taborska Ulica 8, 2000 Maribor, Slovenia

**Keywords:** cell viability, in vitro toxicology, dynamic direct cell death assay

## Abstract

The main focus of in vitro toxicity assessment methods is to assess the viability of the cells, which is usually based on metabolism changes. Yet, when exposed to toxic substances, the cell triggers multiple signals in response. With this in mind, we have developed a promising cell-based toxicity method that observes various cell responses when exposed to toxic substances (either death, division, or remain viable). Based on the collective cell response, we observed and predicted the dynamics of the cell population to determine the toxicity of the toxicant. The method was tested with two different conformations: In the first conformation, we exposed a monoculture model of blood macrophages to UV light, hydrogen peroxide, nutrient deprivation, tetrabromobisphenol A, fatty acids, and 5-fluorouracil. In the second, we exposed a coculture liver model consisting of hepatocytes, hepatic stellate cells, Kupffer cells, and liver sinusoidal endothelial cells to rifampicin, ibuprofen, and 5-fluorouracil. The method showed good accuracy compared to established toxicity assessment methods. In addition, this approach provided more representative information on the toxic effects of the compounds, as it considers the different cellular responses induced by toxic agents.

## 1. Introduction

Cell death is a dynamic process with different pathways (apoptosis, necroptosis, ferroptosis, etc.) that lead to different responses. Each pathway has many different subroutines with many different molecular mechanisms that can overlap [1]. Most subroutines last between 3 and 48 h and usually end in cell death. However, during this period, the cell can reverse the cell death pathway (a process known as anastasis). Anastasis can be triggered at any point along the death pathway until the cell is no longer able to reverse the death pathway [2]. In addition, certain cell death pathways release cytokines that trigger the proliferation of neighbouring cells (of the same or a different cell type). This is a mechanism that allows the tissue to compensate for the death of neighbouring cells and is called compensatory proliferation (a pathway involved in normal homeostatic mechanisms) [3]. Thus, when a toxic agent is injected into a cell population, the cells either die, remain intact or divide [4,5]. 

In this sense, evaluating the toxicity of a compound using a cell model can be misleading if not all the possible outcomes are considered. For example, a toxic compound may induce a decrease in the cell number of a cell population, which may afterwards renew it. Moreover, this cell population may be a functional cell population to the same extent as before the exposure [4,6]. Thus, cell population dynamics change over time. Previous studies explored the idea of using methods that consider the toxicity of compounds over time [7,8]. However, they do not consider cell population dynamics [7], or they use indirect measurements to determine cell viability [8]. This, again, may result in a higher frequency of false positive events, which were observed also in in vitro comet assays [9] and is especially true for cell lines that are histologically and genetically more heterogeneous, like glioblastoma cells [10]. Another study in this cell line also highlighted inter-assay differences, which may be related to the nature of the compounds [11].

To solve this problem, we propose a new method to determine a substance’s toxicity that encompasses the three outcomes mentioned above and measure it via direct determination. Based on these outcomes, we observed and predicted cell population dynamics to determine the toxicity of the toxic agent. We tested the suitability and accuracy of the proposed method with two different conformations to obtain more comprehensive data in regard to its usability. In the first conformation, we exposed a monoculture model of blood macrophages to different toxic agents. In the second, we exposed a coculture liver model to various hepatotoxic compounds. With this experimental design, we tried to observe whether the cell population dynamics would reduce false positive events when assessing cell viability and predict with higher accuracy the toxicity of the compounds on various cell model arrangements. We observed that the proposed method is consistent with the toxicity assessment of other well-established toxicity assessment methods. Moreover, the toxicity assessment of the hepatotoxic drugs with this new method is in line with the available clinical data [12,13,14,15,16].

## 2. Results

### 2.1. Toxicity Assessment of Various Compounds with Different Methods

MTT toxicity assessment was concordant with the cell count method in all the samples of the fatty acids (FA) and hydrogen peroxide (30%) (H_2_O_2_)treatments (Figure 1a). The toxicity assessment with the MTT assay overestimates the toxic effect of the FA and H_2_O_2_ exposure in TLT cells (Figure 1a). A similar trend was observed when depriving the TLT cells of nutrients, but it was not statistically significant.

The propidium iodide (PI) toxicity assessment underestimates the toxicity of FA, H_2_O_2_, tetrabromobisphenol A (TBBPA), and Ultraviolet light (UV) in comparison to MTT and cell count methods (Figure 1a). The toxicity assessment with the PI method and cell count method were similar when starving the cells or treating them with 5-fluorouracil (5-FU) (Figure 1a).

When analysing the cell number with the DD Cell-Tox method in the untreated samples (control), we counted more cells on Day 1 than on Day 0 (Figure 1b) despite the already formed monolayer at Day 0. On Day 1, the dead cells ratio was around 12%, while the ratio of dividing cells was 20% (Table 1). Thus, a pro-proliferative effect was expected on the following days (Day 1+).

Samples treated with FA and H_2_O_2_ had fewer cells on Day 1 than on Day 0 (Figure 1b). On Day 1, both groups did not have any dividing cells, and the dead cell ratio was high (35% and 33% for FA and H_2_O_2_ treatments, respectively) (Table 1). Thus, an anti-proliferative effect was expected (Day 1+), which was confirmed by the analysis with the cell count method on Day 2 (Figure 1c). Indeed, we observed a significantly lower cell number in both treatments when comparing them to the untreated samples.

The samples treated with TBBPA and UV had less cells on Day 1 than on Day 0 (Figure 1b). On Day 1, we observed cell division in 3% and 7% of the samples in TBBPA and UV treatments, respectively (Table 1). In contrast, most of the cells were dead (dying), specifically 45% and 26% in samples treated with TBBPA and UV treatments, respectively (Table 1). Thus, an overall anti-proliferative effect is expected in both treatments (Day 1+), as was observed by the cell count method on Day 2 (Figure 1c). These differences in cell number between the treated and untreated samples were statistically significant.

We counted fewer cells on Day 1 than on Day 0 in samples treated with nutrient starvation and 5-FU (Figure 1b). On Day 1, 26% and 39% cell division ratios were determined in 5-FU and nutrient starvation treatments, respectively (Table 1), while the cell death ratio in both treatments was lower than the cell division (7% and 12% in 5-FU and nutrient starvation, respectively) (Table 1). Thus, a pro-proliferative effect is expected (Day 1+), which was corroborated by the cell count method on Day 2 (Figure 1c).

The toxicity assessment (given as cell number predictions) with the DD Cell-Tox method at Day 1+ (after the treatment) (Figure 1b) for the coming days was similar to the cell number counted at Day 2 with the Cell Count method (Figure 1c). It is noteworthy that the DD Cell-Tox method accurately predicts which treatments had an overall pro-proliferative effect and anti-proliferative effect. This is visible as the cell number prediction on Day 1+ (Figure 1b) and was then corroborated by the cell count method on Day 2 (Figure 1c).

### 2.2. Toxicity Evaluation of DD Cell-Tox Method in Complex Cell-Based Liver Model in Comparison to Clinical Data

The cell number increased in all samples of the liver model (Table 2). We did not observe any toxic response in the samples treated with rifampicin (RIF), ibuprofen (IBU), or 5-FU (Figure 2a). However, the prediction of the cell number infers that the cell number may be reduced in the future. This is consistent with the observations made in patients treated with rifampicin, ibuprofen, or 5-FU (Figure 2b) [12,13,14,15,16]. In the first week of treatment, no toxic effect was observed in the liver of the patients. In the following weeks, however, the patient’s liver markers rose above normal values, indicating a possible toxic effect on the liver (Figure 2b).

## 3. Discussion

Cells are constantly dying and reproducing to maintain the homeostasis of the organ. Thus, assessing the viability of a cell population at a given time may not describe the dynamics of the cell population (in terms of its viability) [4,5]. This is especially true when inoculating a toxic substance. A group of cells that is categorised as dead may reverse course and even induce neighbouring cells to proliferate. The frequency of these events (known as false positives) could be lower if viability is determined using a method that considers the dynamics of the cell population. To estimate the imminent dynamics of a cell population (t_1+h_), we can calculate the difference in cell number between two different time points (t_0_ and t_1_), count the dividing cells, and subtract the dead cells at time point t_1_ [Equation (1)].
(1)$$Cell\;number_{t_{1}+h}=Cell\;number_{t1}-Cell\;number_{t_{0}}+Dividing\;cells_{t1}-Dead\;cells_{t1}$$

Most cell death pathways last 48 h [1,2,17,18], and most cells divide for 24 h [19]. In addition, tissue-repairing signals that trigger proliferation are initiated approximately 6 h after the onset of programmed cell death [3]. Therefore, 24–48 h after inoculation with the toxic compound, the majority of cells would either die, maintain their integrity, or divide.

Previous studies observed that the exposure of a toxic compound to a cell population can be accompanied not only by cell death but also by cell proliferation (known as compensatory proliferation) [20,21]. This situation can be even more pronounced in heterogeneous cell populations; certain cell types can be induced to die while other cell types can be induced to proliferate [20,21]. Complex models such as the liver can be particularly striking, where dying hepatocytes (which make up 80% of the liver) can induce fibroblast proliferation through compensatory proliferation [22,23]. If these assumptions are contemplated, the toxicity of a given compound can be determined with greater accuracy than existing methods because all possible outcomes are considered (in contrast to existing methods). In this sense, the proposed method predicted cell population dynamics in two models (monoculture of TLT and complex liver model with four different cell types) with high accuracy.

When exposing the TLT cells to 12.5 mM H_2_O_2_, we expected to observe a higher frequency of necroptotic cells [24]. Thus, we expected the inhibition of caspase 8 and the formation of a receptor-interacting protein kinases 1 and 3 (RIP1/RIP3) necrosome that phosphorylates mixed-lineage kinase domain-like pseudo kinase (MLKL), which initiates the characteristic membrane rupture [25,26]. On the other hand, 266 µM TBBPA or 500 µM 5-FU exposure should induce apoptosis on TLT cells [27,28]. In this case, the executioner caspase 3 is needed for the completion of the apoptotic pathway, regardless of the downstream molecules, for caspase 8 in the case of extrinsic apoptosis or for mitochondrial outer membrane permeabilization (MOMP) in the case of the intrinsic apoptosis [1]. Cell death (mainly by autophagy) was also induced by nutrient starvation [29], characterized by cytoplasmic vacuolization, which is phagocytised and subsequently degraded by the lysosomes [1]. For this purpose, 50 microliters of the cell medium were added, and the medium was not replenished for 72 h. To induce ferroptosis (via lipid peroxidation) [30], 15 µL of FA was added to the cell medium. Ferroptosis is initiated via oxidative perturbations within the cell and mainly regulated by glutathione peroxidase 4 (GPX4) [1]. Finally, cell death (mainly parthanatos and apoptosis) was also induced by the UV irradiation of the cells for 5 min [31]. In both cases (either apoptosis or parthanatos) the hyperactivation of a specific component of the DNA damage response poly(ADP-ribose) polymerase 1 (PARP1) may be the trigger [1]. After the exposure to the above-mentioned toxic agents, we observed a reduction in cell viability in all cases, which was expected [24,27,28,29,30,31]. At this point, we replenish the media in all samples with a cell culture medium without the toxic substances. At a time of 24 h after changing the media, the cells exposed to nutrient deprivation and 500 µM 5-FU proliferated, while the remaining cells showed reduced viability. The nutrient deprivation triggers an autophagy response [29], which we observed in the first 24 h (Figure 1b, Day 1). However, as previously observed by other studies [32,33,34], after the cells were supplied with new nutrients, they changed this pathway and favoured cell division (Figure 1b, Day 1+). 5-FU induces cell death arrest in cells [35] and, subsequently, cell death, as we observed in the samples treated with it (Figure 1b, Day 1). However, after changing the medium, the cells reversed the cell death pathway and induced proliferation (Figure 1b, Day 1+). As a result, the cells showed a higher cell number 24 h after cell renewal (Figure 1c, Day 2). As observed in a previous study, cells can re-enter the cell cycle after cell cycle arrest [36]. In another study, the differences in mechanisms between cell cycle arrest switching and proliferation were also discussed (growth factors and the focal adhesion kinase activation of the cell cycle and the arrest induced by yes-associated protein phosphorylation) [37]. Once the cell cycle arrest molecule is washed away, cells can resume or continue the cell cycle and proliferate [37].

It is noteworthy that the untreated sample in the TLT cell experiment showed a relatively high ratio of cell death to cell division. However, considering that the blood macrophages formed a monolayer at the beginning of the experiment and had a doubling time of 20 h, further TLT cultivation is expected to induce them to die and proliferate [38].

We also tested the DD Cell-Tox method in a liver model with four different types of liver cells (LSEC, Kupffer, HSC, and hepatocytes). We exposed the model to three different hepatotoxic compounds and evaluated the cell population dynamics using the DD Cell-Tox method. The cell population predictions were consistent with the clinical data (Figure 2b) [12,13,14,15,16].

RIF is an antibiotic that is commonly used to treat tuberculosis. Treatment with RIF can lead to toxicity after the first week of exposure. The mechanism of hepatotoxicity is related to the induction of various cytochrome P450 (CYP) enzymes, including CYP3A4 and ABC C2. The toxicity of the resulting metabolites may be the main cause of toxicity [12]. Further studies also point out that the hepatic toxicity mechanisms of RIF may be related to the upregulation of bile acid transporters, including multidrug resistance-associated proteins 2, 3, and 4. Moreover, RIF inhibits the synthesis of uridine diphosphate glucuronosyltransferase 1a1, which reduces the metabolism of toxic components [39]. In this sense, no toxic response was observed during the first week, as documented in clinical data (Figure 2b) [15].

IBU is a non-steroidal anti-inflammatory drug (NSAID) that is used to treat inflammation and mild pain in patients. IBU competitively inhibits both cyclooxygenase 1 and 2 enzymes, leading to decreased prostaglandin synthesis, which causes renal vasoconstriction. Additionally, IBU can prolong bleeding time by inhibiting thromboxane A2. IBU can have potentially hepatotoxic effects with prolonged constant use (over a week of constant use at high doses), possibly due to the accumulation of toxic metabolic by-products [40]. During the first week of exposure, we did not observe any toxic response, as it happened with the clinical reference we found (Figure 2b) [16].

5-FU is used as a cytotoxic anticancer agent for the treatment of various types of cancer (especially in the colon). 5-FU has been associated with clinically apparent acute liver injury in certain cases. Clinical hepatotoxic markers increase slightly when patients are treated with 5-FU. This is mainly due to the accumulation of toxic metabolic by-products and the inhibition of thymidylate synthase [13,41,42,43]. Moreover, the downregulation of the transcription factor E2F1 and the protein Echinoderm microtubule-associated protein-like 2, as well as the decrease in TCA cycle intermediates, disables cell growth [44]. All the previous molecular disruptions may induce a cell population reduction in comparison to the untreated sample during the first week of exposure (Table 2). Longer exposure to 5-FU would have a significant toxic effect on the liver model, as observed in clinical references (Figure 2b) [14].

Thus, the method is compatible and accurate with a complex model with different cell types. In addition, this method allows a more informative observation of potential cell replication and cell death events. With the DD Cell-Tox method, we can observe cell death or proliferation activity in the first 24 h and predict possible cell deaths and cell doublings in the next 24 h. In this way, we cover a 48 h window, which is normally the longest period for programmed cell death or cell doubling.

Limitations

The goal of this manuscript was to develop a new cell viability method based on the direct determination of cell viability, describe its protocol, and test its accuracy in different situations. We also consider the various cell responses when treated with toxic compounds. In this sense, all the cell death pathways that surpass the point of no return will result in cell death, while dividing cells will divide, regardless of the pathway. Yet, the next step should include the use of Western blotting or ELISA to measure key proteins involved in apoptosis, necrosis, and survival pathways (e.g., caspases, Bcl-2 family proteins, p53). By doing so, we may relate the parameters of the method with the different cell death pathways. Under these premises, live imaging could provide a more comprehensive knowledge of the specific cell death pathways, especially, but not limited to, the data related to the morphological changes induced by each cell death pathway.

At a certain point, the cell can initiate the cell death process, but the test does not classify it as dead, because the test only determines the cells after the “point of no return”. However, counting a cell that is dead before the point of no return can lead to a false positive result, as the cell can reverse the process.

The method is based on the cell membrane potential, which may make the viability assessment difficult in cells that have a greater depolarization capability, like neurons or myocytes. Thus, special effort is needed to discern between transient and permanent cell membrane depolarization. 

Likewise, it should be interesting to test the method with toxins that temper the cell membrane transport channels, like tetrodotoxin or saxitoxin. The ability of these toxins to bind to ion channels should affect the cell membrane potential and may result in false positive events. Morphological as well as molecular studies when performing this method should aid in understanding the parameters and time at which the cells are dead. 

Innovative methods for testing carcinogenic cells continue to be a focus of research and development in oncology and cancer biology [45,46]. Cancer treatments aim to preserve the function of healthy cells by maintaining homeostasis while inducing the cell death of cancer cells to control the disease [47]. This method is of particular interest for research with carcinogenic cells for use in cancer models, as it assesses cell death and proliferation.

## 4. Materials and Methods

### 4.1. Cells

Human blood macrophages called TLT (CVCL_6C16), a spontaneously immortalized cell line previously isolated and characterized at the University of Maribor (Slovenia), were used. TLT cells were cultured in colourless Williams E medium (Thermo Fisher Scientific, Waltham, MA, USA) supplemented with 5 wt.% foetal bovine serum (Gibco, Thermo Fisher Scientific, Waltham, MA, USA). L-glutamine (2 mM, Sigma-Aldrich, San Luis, MO, USA), penicillin (100 U mL^−1^, Sigma-Aldrich), and streptomycin (1 mg/mL^−1^, Fluka, Buchs, Switzerland) were also added. Cells were cultured in 25 cm^2^ culture flasks (Corning, New York, NY, USA) at 37 °C and 5% CO_2_.A liver model was built by coculturing liver sinusoidal endothelial cells (LSEC), human stellate cells (HSC), Kupffer cells (nonparenchymal liver cells, NPC) from ZEN-BIO (Durham, NC, USA), and hepatocytes from Lonza (Basel, Switzerland). NPC and hepatocytes were grown separately in a 25 cm^2^ flask (NUNC, Roskilde, Denmark) in a controlled environment at 37 °C and 5% CO_2_ and later seeded together in a 96 well microplate (NUNC, Denmark) to build the liver model.

### 4.2. Chemicals

Chemical 3-(4,5-dimethylthiazole-2-yl)-2,5-phenyl tetrazolium bromide (MTT reagent) was acquired from Sigma-Aldrich (Saint Louis, MO, USA). Phosphate-buffered saline solution (PBS), FluoVolt™ (FV) Membrane Potential Kit, Vybrant™ DyeCycle™ Ruby Stain (Vybrant), and PI were purchased from Thermo Fisher Scientific (Waltham, MA, USA).

H_2_O_2_, tetrabromobisphenol A, RIF, IBU, and 5- FU were purchased from Merck (Merck KGaA, Darmstadt, Germany). Fatty acids (FA) were obtained from oil digestion with lipase (Merck KGaA, Germany).

### 4.3. Protocol

The protocol of the DD Cell-Tox method is the same as the protocol described in Appendix A.3. in the manuscript “A Promising Method for the Determination of Cell Viability: The Membrane Potential Cell Viability Assay” [48] (Figure 3a,b). As described in this manuscript, the cells with FV intensity outside the “viable FV intensity range” (a term described in the manuscript) were categorised as dead. In contrast, the cells with FV intensity within the “viable FV intensity range” were categorised as viable cells. However, in the DD Cell-Tox method, we count the number of dividing cells. The cells that were temporally depolarized and had a higher DNA content (2n) were categorised as duplicating cells [48]. By doing so, we were able to observe and predict the cell number in the following days by calculating the cell number at the time of the experiment, counting the dividing cells, and subtracting the dead cells at the time of measurement. This method provides the cell dynamics, and the toxicity of the drug is determined by it (Figure 3c). In the Result Section, we noted the projections of the cell population dynamics with a + sign. Thus, the day 1+ and day 5+ results depicted with an arrow are the cell population dynamics we expected in the following days.

### 4.4. Testing the Method in the First Configuration (Monoculture)

We seeded TLT cells in two 96 well microplates and waited until they reached 100% confluency. After cell monolayer formation, the cells on a 96-well microplate were treated with different toxic substances 12.5 mM H_2_O_2_, 266 µM TBBPA, 500 µM 5-FU, nutrient deprivation, and 15 µL FA for 24 h. We exposed the cells seeded in the other 96 well microplate to UV light for 5 min. At this point, we tested the viability of the samples with three different methods: DD Cell-Tox method, PI method [49], and MTT assay [50] (Figure 4c). After the treatments, we renewed the cell medium in all samples with a new cell culture medium without the toxic agents. We tested the viability of the samples with the cell count method [51] (Figure 4d) and compared its toxicity assessment with the DD Cell-Tox method, PI method, and MTT assay (where we normalized the results of treated samples to the untreated sample) (Figure 4e). Note that the viability assessment with the DD Cell-Tox method also produces a projection of the cell population dynamics, denoted by the nomenclature of day 1+ (Figure 4e).

### 4.5. Testing the Method in the Second Arrangement (Coculture Model)

We cultured the liver cells separately until confluency was reached and then reseeded them to build the liver model. We seeded 20,000 hepatocytes and 5000 NPCs (Figure 5a). We cultured them with Hep medium (as described in the manuscript “In Vitro Human Liver Model for Toxicity Assessment with Clinical and Preclinical Instrumentation” [52]). We individually added three different hepatotoxic drugs to the cell culture media at the following final concentrations: 50 µmol/L RIF, 1 mmol/L IBU, or 500 µmol/L 5-FU. Two days after the addition, we renewed the same culture medium (Figure 5c). Five days after the initial exposure, we measured the viability of the cells with the DD Cell-Tox method and compared the measured value to the clinical data obtained from the literature [12,13,14,15,16] (Figure 5d). Note that the viability assessment with the DD Cell-Tox method also produces a projection of the cell population dynamics, as denoted by the nomenclature of day 5+ (Figure 5d).

### 4.6. Statistical Analysis

We used the ANOVA Tukey’s HSD test to compare the viability assessments of the cell count, MTT, and PI methods. We used Mann–Whitney to test the difference between the cell numbers obtained through the DD Cell-Tox and cell count methods.

## 5. Conclusions

Cell viability tests should determine cell viability as a changing parameter and not as a single moment. Based on this basic observation regarding the measurement of viability, we recommend a dynamic approach to measure an observed cell population rather than cell viability at a specific point in time. This is particularly important when evaluating the effect of potential anticarcinogenic compounds, where different cell responses are expected in cancerogenic and healthy cells after their exposure to them. The DD Cell-Tox method incorporates these premises and has been successfully tested in different situations. However, many alternative tests should be performed to evaluate the potential broad application of the method.

## Figures and Tables

**Figure 1 ijms-25-05133-f001:**
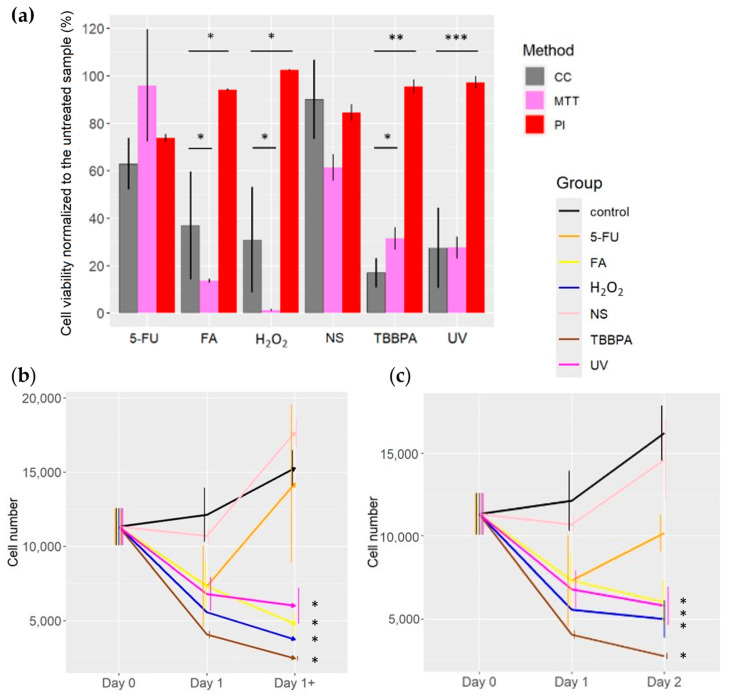
Viability assessment of the various methods in comparison to the cell count method. (**a**) Viability assessment normalized to the untreated sample comparison between Cell count (CC), MTT (MTT), and PI methods (PI). (**b**) Viability assessment with the DD Cell-Tox method (the arrows represent the cell population dynamics in the following days, with the nomenclature of day 1+). (**c**) Cell count of the samples at different time points. (>0.05, * 0.05, ** 0.01, *** 0.001). CC: Cell count method; MTT: MTT assay; PI: PI method; 5-FU: 5-fluorouracil; FA: Fatty acids; H_2_O_2_, NS: Nutrient starvation; TBBPA: tetrabromobisphenol A; UV: Ultraviolet light.

**Figure 2 ijms-25-05133-f002:**
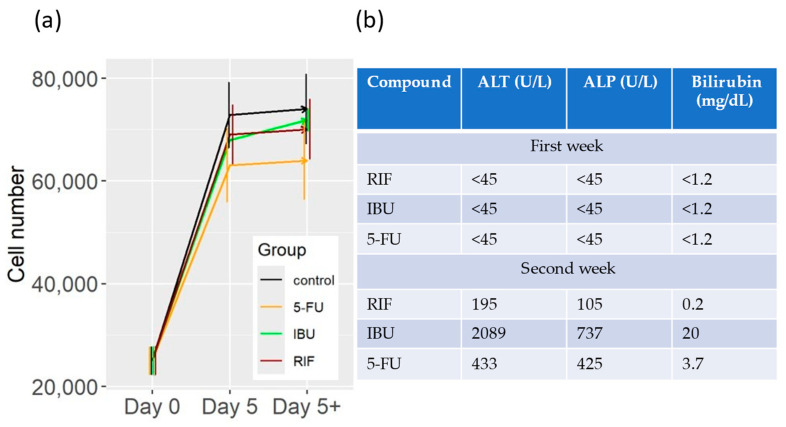
Comparison between the DD Cell-Tox method toxicity assessment in a liver model and the existing clinical data for RIF, IBU, and 5-FU. (**a**) DD Cell-Tox method toxicity evaluation of RIF, IBU, and 5-FU (the arrows represent the cell population dynamics in the following days, with the nomenclature of day 5+). (**b**) ALT, ALP, and bilirubin values of patients treated with RIF, IBU, and 5-FU. 5-FU: 5-fluorouracil, IBU: ibuprofen, RIF: rifampicin.

**Figure 3 ijms-25-05133-f003:**
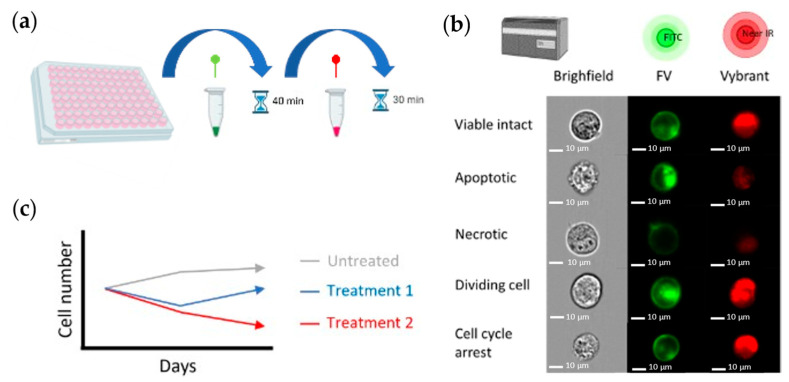
DD Cell-Tox method toxicity assessment scheme. (**a**) The method’s protocol. (**b**) Cell response measurement with the imaging flow cytometer MK2 (ISX). (**c**) Diagram (analysis) of the cell dynamics for the toxicity assessment of substances.

**Figure 4 ijms-25-05133-f004:**
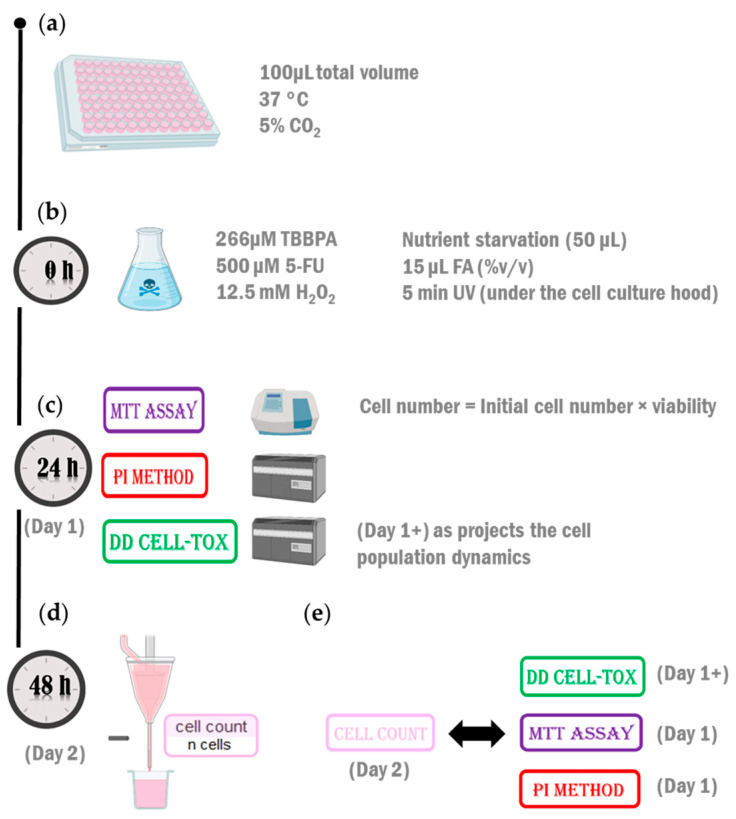
The experimental setup for DD Cell-Tox accuracy assessment in blood macrophages. (**a**) Cells were seeded and cultured until 100% confluency was achieved. (**b**) After cell attachment, the samples were treated with 12.5 mM H_2_O_2_, 266 µM TBBPA, 500 µM 5-FU, 15 µL (%*v*/*v*) FA, nutrient starvation, and 5 min exposure to UV. (**c**) At a time of 24 h after the treatment, cell viability was assessed with the MTT assay and the PI method. At the same time, the DD Cell-Tox method was performed. The formula to calculate the cell number prediction of the population is depicted in the graph below the formula used for the calculation. (**d**) At a time of 48 h after the treatment, cell viability was assessed with the cell count method. (**e**) The accuracy of the cell count method 48 h after treatment was compared to the DD Cell-Tox method, MTT assay, and PI method. 5-FU: 5-fluorouracil; FA: Fatty acids; H_2_O_2_; NS: Nutrient starvation; TBBPA: tetrabromobisphenol A; UV: Ultraviolet light.

**Figure 5 ijms-25-05133-f005:**
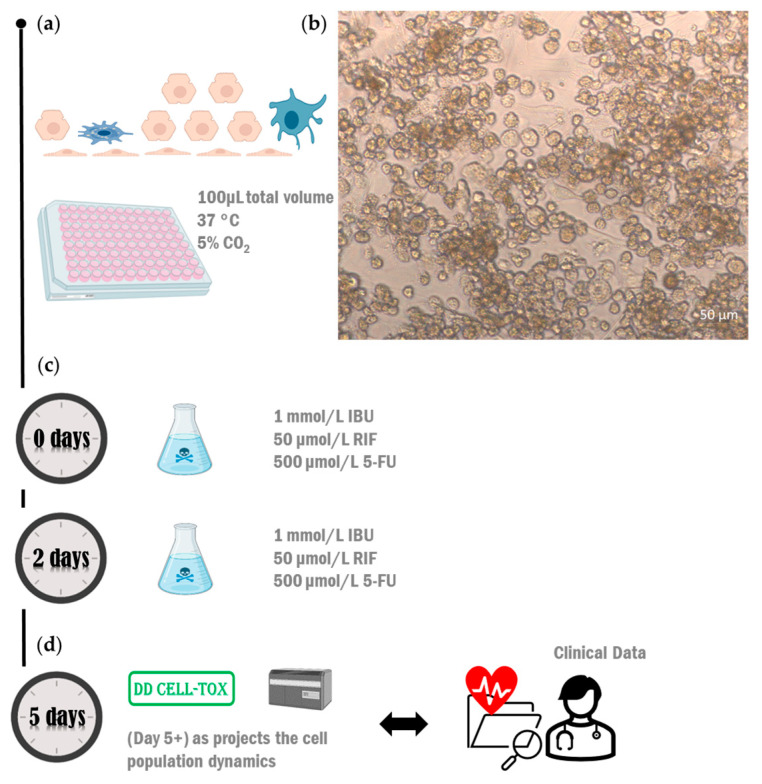
The experimental setup for DD Cell-Tox accuracy assessment in a cell-based liver model. (**a**) We designed a liver model with four cell types: LSEC, HSC, Kupffer cells, and hepatocytes. (**b**) Brightfield image at 20× magnification obtained with an inverted microscope (Axiovert 40 CFL, ZEISS, Jena, Germany). (**c**) We exposed the liver model to 1 mmol/L IBU, 50 µmol/L RIF, or 500 µmol/L 5-FU for 5 days (we renewed the medium after 2 days). (**d**) We measured the cell viability with the DD Cell-Tox method. 5-FU: 5-fluorouracil; IBU: ibuprofen; RIF: rifampicin.

**Table 1 ijms-25-05133-t001:** Summarizing the effect of contaminants in TLT cells.

Group	Number of Initial Cells	Number of Final Cells	Dead Cells (%)	Division Cells (%)
Control	11,329	12,130	12	20
5-FU	11,329	7327	7	26
FA	11,329	7318	35	0
H_2_O_2_	11,329	5564	33	0
NS	11,329	10,694	12	39
TBBPA	11,329	4064	45	3
UV	11,329	6797	26	7

All the treatments lasted 24 h. 5-FU: 5-fluorouracil; FA: Fatty acids; H_2_O_2_; NS: Nutrient starvation; TBBPA: tetrabromobisphenol A; UV: Ultraviolet light.

**Table 2 ijms-25-05133-t002:** Summary of the DD Cell-Tox method toxicity evaluation of three well-known hepatotoxic drugs.

Group	Number of Initial Cells	Number of Final Cells	Cell Number Increase (%)	Dead Cells (%)	Division Cells (%)
Control	25,000	72,797	291	3	2
5-FU	25,000	63,056	252	4	3
IBU	25,000	67,919	272	3	4
RIF	25,000	68,995	276	2	2

All the treatments last 5 days. 5-FU: 5-fluorouracil, IBU: ibuprofen, RIF: rifampicin.

## Data Availability

The authors agree to make data and materials supporting the results or analyses presented in this paper available upon reasonable request.

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
