# Peer review of "Finding a Direct Method for a Dynamic Process: The DD (Direct and Dynamic) Cell-Tox Method"

_ijms, 2024, doi:10.3390/ijms25105133_

Round 1
Reviewer 1 Report
Comments and Suggestions for Authors
In this study, the authors developed a promising cell-based toxicity method that observes various cell responses when exposed to toxic substances (either death, division or remaining viable). The authors recommend a dynamic approach to measure an observed cell population rather than cell viability at a specific point in time.
Comments
This is an interesting study. The reviewer has some concerns as follows:
1. There seems to be a similar testing method currently, such as Joeng, L., et al. Validation of the dynamic direct exposure method for toxicity testing of diesel exhaust in vitro. ISRN Toxicol. 2013 Aug 5;2013:139512. doi: 10.1155/2013/139512; Vaidya, T.R., et al. A three-dimensional and dynamic (3DD) cell culture system for evaluation of pharmacokinetics, safety and efficacy of anti-cancer drugs. Curr Pharmacol Rep. 5, 460–467 (2019). https://doi.org/10.1007/s40495-019-00198-1. How does the current method differ from other methods? It can be explained or discussed.
2. In Figures 1 and 5, this is a bit unclear about day1+ / day5+. Do day1+ / day 5+ and day2 / day6 mean the same thing?
3. In Figure 1a, there seems to be no normal control group. Moreover, how long is the drug treatment period?
4. In Tables 1 and 2, please indicate the duration for drug treatment.
5. In Table 1 title, “….in TLT” can be changed to “…in TLT cells”.
Author Response
- There seems to be a similar testing method currently, such as Joeng, L., et al. Validation of the dynamic direct exposure method for toxicity testing of diesel exhaust in vitro. ISRN Toxicol. 2013 Aug 5;2013:139512. doi: 10.1155/2013/139512; Vaidya, T.R., et al. A three-dimensional and dynamic (3DD) cell culture system for evaluation of pharmacokinetics, safety and efficacy of anti-cancer drugs. Curr Pharmacol Rep. 5, 460–467 (2019). https://doi.org/10.1007/s40495-019-00198-1. How does the current method differ from other methods? It can be explained or discussed.
We thank the reviewer for the effort of finding methods that may be similar to ours, so we may compare to them. In the first method developed by Joeng, L. et al. the title of the manuscript seems similar to our method. Yet, the term “direct” in their case refers to the direct exposure of the toxic compounds, while we use the term “direct” to describe the methodology of our method to determine the viability of the cell. On the contrary, the method developed by Joeng, L. et al. that is based on metabolic activity, thus, indirect tests to determine the viability of the cell. Moreover, they don’t consider the population dynamics, they rather measure the viability of the cells at two different points to determine any delayed cytotoxic effects. Likewise, the method developed by Vaidya et al. does not measure cell viability directly but rather observed caspase markers to assess cell viability.
- In Figures 1 and 5, this is a bit unclear about day1+ / day5+. Do day1+ / day 5+ and day2 / day6 mean the same thing?
We understand the confusion and thank the reviewer for showing the issue. In the case of the days+, they are projections of the cell population based on the formula we described in Equation 1. In this sense, we compared the projection of day 1+ with the cell number on day 2, as measured by the cell count method. We also compared the projection of day 5+ with the clinical data obtained from patients treated with the same drugs. We implemented these changes in the manuscript and changed the figures accordingly.
- In Figure 1a, there seems to be no normal control group. Moreover, how long is the drug treatment period?
We apologize for not stating that we normalized the viability measurements in Figure 1a with the untreated groups of each test. That way, we harmonized the endpoints of each test to facilitate the comparison between them. In the case of the section “2.1. Toxicity assessment of various compounds with different methods.”, the drug treatment period was 24 hours. In the case of the section “2.2. Toxicity evaluation of the DD Cell-Tox method in a complex cell-based liver model in comparison to clinical data,” the drug treatment was 5 days.
- In Tables 1 and 2, please indicate the duration for drug treatment.
We implemented the changes in the manuscript as suggested.
- In Table 1 title, “….in TLT” can be changed to “…in TLT cells”.
We implemented the changes in the manuscript as suggested.
Reviewer 2 Report
Comments and Suggestions for Authors
1. The description of the experimental design lacks detail in several areas, particularly concerning the configuration of cell culture conditions and the specific concentrations of the compounds used.
2. Use Western blotting or ELISA to measure key proteins involved in apoptosis, necrosis, and survival pathways (e.g., caspases, Bcl-2 family proteins, p53) following treatment with toxicants.
3. Utilize advanced imaging techniques such as time-lapse microscopy to observe morphological changes in cells in real-time after exposure to toxicants. This can provide visual confirmation of processes like apoptosis, necrosis, or autophagy.
4. Although the paper discusses the outcomes of toxicant exposure, it could delve deeper into the molecular pathways and mechanisms by which these toxicants affect cell viability. This could include specific signaling pathways, gene expression changes, or protein interactions.
5. Expanding on the limitations section by discussing potential biases, the scope of applicability of the method (e.g., types of cells, toxicants), and future improvements would provide a more balanced view and suggest avenues for further research.
6. Provide more background on the different types of cell death (apoptosis, necroptosis, ferroptosis, etc.), highlighting their unique features and implications in toxicity studies. This could help readers understand the complexity and relevance of studying multiple pathways.
7. Strengthen the justification for your study by linking the introduction more directly to your research objectives. Explain why your method (DD Cell-Tox) could be a significant improvement over current methods, considering the dynamics of cell populations under toxic stress.
8. End the introduction with a clear statement of your hypothesis and the objectives of your research. This sets the stage for the "Materials and Methods" section and guides the reader through the rationale behind your experimental design.
Author Response
- The description of the experimental design lacks detail in several areas, particularly concerning the configuration of cell culture conditions and the specific concentrations of the compounds used.
We added new figures and added more details regarding the cell culture conditions and the specific concentrations of the compounds in the revised version. We want to point out that the purpose of this study was to use graphs and figures to illustrate pertinent data for all variables on the specific concentrations of the compounds and the conditions of the cell culture.
- Use Western blotting or ELISA to measure key proteins involved in apoptosis, necrosis, and survival pathways (e.g., caspases, Bcl-2 family proteins, p53) following treatment with toxicants.
The goal of this article is to comprehend the basic idea behind a compound's toxic effects. The many types of programmed cell death and the conventional and non-conventional methods to identify them have not been examined. We agree that the suggested approach needs to be addressed for the implementation of the method, and we included it in the limitation section. Once the accuracy and validity of the method are proven, its versatility needs to be tested.
- Utilize advanced imaging techniques such as time-lapse microscopy to observe morphological changes in cells in real-time after exposure to toxicants. This can provide visual confirmation of processes like apoptosis, necrosis, or autophagy.
We link this point to the prior, as we think they are related. So, we agree that live imaging will enable the determination of the cell death pathway. However as explained above, we think this is not within the scope of this manuscript. Nevertheless, we greatly appreciate the suggestion, and we think it’s very interesting, so we will observe the morphology changes in the future to determine the cell death pathway.
- Although the paper discusses the outcomes of toxicant exposure, it could delve deeper into the molecular pathways and mechanisms by which these toxicants affect cell viability. This could include specific signaling pathways, gene expression changes, or protein interactions.
As the reviewer points out in the second point, there are key proteins that may help determine the specific cell death pathway. With this information, we could comprehensively analyse the molecular pathways and mechanisms by which these toxicants affect cell viability. Without this information, we cannot discuss with the required level of certainty the specific signalling pathways, gene expression changes, or protein interactions. Nevertheless, we discussed the potential molecular pathways and mechanisms the toxic compounds may induce in cells, since this must be the main topic of the next manuscript related to the method.
- Expanding on the limitations section by discussing potential biases, the scope of applicability of the method (e.g., types of cells, toxicants), and future improvements would provide a more balanced view and suggest avenues for further research.
We agree that the limitations section falls short of discussion. We implemented the suggestions of the reviewer in this section, as well as other future tests that may improve the implementation of the method and future validation.
- Provide more background on the different types of cell death (apoptosis, necroptosis, ferroptosis, etc.), highlighting their unique features and implications in toxicity studies. This could help readers understand the complexity and relevance of studying multiple pathways.
We agree with the reviewer comment, and we implement the information regarding the different apoptotic pathways in the manuscript.
- Strengthen the justification for your study by linking the introduction more directly to your research objectives. Explain why your method (DD Cell-Tox) could be a significant improvement over current methods, considering the dynamics of cell populations under toxic stress.
We thank the reviewer for the comment , we think this goes in line with the comments from the first Reviewer. In the revised manuscript we think that the suggested change helped to put into perspective the purpose of the method.
- End the introduction with a clear statement of your hypothesis and the objectives of your research. This sets the stage for the "Materials and Methods" section and guides the reader through the rationale behind your experimental design.
We ended the introduction as the reviewer suggested, and we hope that the transition from the introduction to the “Materials and Method” section is more seamless than in the previous version of the manuscript. With this experimental design, we tried to observe whether the cell population dynamics would reduce false-positive events when assessing cell viability and predict with higher accuracy the toxicity of the compounds in various cell model arrangements.
Reviewer 3 Report
Comments and Suggestions for Authors
The manuscript titled “Finding a Direct method for a Dynamic process: the DD (Direct and Dynamic) Cell-Tox method” aimed to develop cell-based toxicity method that observes various cell responses when exposed to toxic substances testing the method with two different conformations being a monoculture model and a coculture liver model. Based on the obtained results the method showed good accuracy compared to established toxicity assessment methods providing more representative information on the toxic effects of the compounds as it considers different cellular responses induced by toxic agents.
Authors could address the issue of false positive results due to cytotoxicity and cell death for the subsequent genotoxicity testing (e.g. for the comet assay) since high concentrations of test agents may cause cytotoxicity or cell death, which may give rise to false positive results in the comet assay for instance.
Azqueta A, Stopper H, Zegura B, Dusinska M, Møller P. Do cytotoxicity and cell death cause false positive results in the in vitro comet assay? Mutat Res Genet Toxicol Environ Mutagen. 2022; 881: 503520. doi: 10.1016/j.mrgentox.2022.503520.
Authors could also discuss the reliability of in vitro IC50 values since these values may vary with different cytotoxicity assays. There are studies suggesting that discrepancies exist among in vitro cytotoxicity methods resulting in unreliable drug toxicity profiles being particularly critical for cell lines that are histologically and genetically more heterogeneous.
Damiani E, Solorio JA, Doyle AP, Wallace HM. How reliable are in vitro IC50 values? Values vary with cytotoxicity assays in human glioblastoma cells. Toxicol Lett. 2019; 302: 28-34. doi: 10.1016/j.toxlet.2018.12.004.
Interesting results were obtained for human glioblastoma cells, which are histologically and genetically heterogeneous when treated with more complex natural compound (e.g. bee venom) and afterward evaluated using three different cytotoxicity assays (MTT, Cristal violet, and Trypan blue exclusion assay) indicating also inter-assay differences, especially at higher concentrations tested. Could also the nature of the compound play a role in such a response?
Gajski G, Čimbora-Zovko T, Rak S, Osmak M, Garaj-Vrhovac V. Antitumour action on human glioblastoma A1235 cells through cooperation of bee venom and cisplatin. Cytotechnology. 2016; 68(4): 1197-205. doi: 10.1007/s10616-015-9879-4.
Please enter ethics approval for the blood donors in the Materials and Methods section (4.1. Cells).
Minor remarks:
Page 1 – remove … after (apoptosis, necroptosis, ferroptosis…) and use etc. instead
Pay attention to subscripts and superscripts (e.g. H2O2, CO2, etc.)
Justify the text in all the paragraphs
Author Response
Authors could address the issue of false positive results due to cytotoxicity and cell death for the subsequent genotoxicity testing (e.g. for the comet assay) since high concentrations of test agents may cause cytotoxicity or cell death, which may give rise to false positive results in the comet assay for instance.
Azqueta A, Stopper H, Zegura B, Dusinska M, Møller P. Do cytotoxicity and cell death cause false positive results in the in vitro comet assay? Mutat Res Genet Toxicol Environ Mutagen. 2022; 881: 503520. doi: 10.1016/j.mrgentox.2022.503520.
We agree with the reviewer that the inclusion of this topic increases the quality of the manuscript, and we included it in the introduction section. We think that this is a good introduction as to why we decided to develop this method in the first place.
Authors could also discuss the reliability of in vitro IC50 values since these values may vary with different cytotoxicity assays. There are studies suggesting that discrepancies exist among in vitro cytotoxicity methods resulting in unreliable drug toxicity profiles being particularly critical for cell lines that are histologically and genetically more heterogeneous.
Damiani E, Solorio JA, Doyle AP, Wallace HM. How reliable are in vitro IC50 values? Values vary with cytotoxicity assays in human glioblastoma cells. Toxicol Lett. 2019; 302: 28-34. doi: 10.1016/j.toxlet.2018.12.004.
We thank the reviewer for the comment and we implemented it in the introduction section, since goes in line with the manuscript, which tries to change the endpoint of the cytotoxicity assays.
Interesting results were obtained for human glioblastoma cells, which are histologically and genetically heterogeneous when treated with more complex natural compound (e.g. bee venom) and afterward evaluated using three different cytotoxicity assays (MTT, Cristal violet, and Trypan blue exclusion assay) indicating also inter-assay differences, especially at higher concentrations tested. Could also the nature of the compound play a role in such a response?
Gajski G, Čimbora-Zovko T, Rak S, Osmak M, Garaj-Vrhovac V. Antitumour action on human glioblastoma A1235 cells through cooperation of bee venom and cisplatin. Cytotechnology. 2016; 68(4): 1197-205. doi: 10.1007/s10616-015-9879-4.
We agree with the reviewer that the nature of the compounds affects the inter-assay differences. This was the main reason why we decided to develop this new method, after observing significant differences in the toxicity assessment of emerging compounds during our involvement in the ECSafeSEAFOOD project.
Please enter ethics approval for the blood donors in the Materials and Methods section (4.1. Cells).
We thank the reviewer for pointing out this issue. We didn’t explain as we should the acquisition of this cell line. Our institution was responsible for the isolation and characterization of the cell line which spontaneously immortalized (you may find the reference here https://www.cellosaurus.org/CVCL_6C16). Thus, we work with an established cell line and there is no need for the ethics approval statement.
Minor remarks:
Page 1 – remove … after (apoptosis, necroptosis, ferroptosis…) and use etc. instead
We corrected the issue in the manuscript.
Pay attention to subscripts and superscripts (e.g. H2O2, CO2, etc.)
We corrected the issue in the manuscript.
Justify the text in all the paragraphs
We corrected the issue in the manuscript.
Round 2
Reviewer 1 Report
Comments and Suggestions for Authors
This manuscript has a great improvement and can be accepted.
Reviewer 2 Report
Comments and Suggestions for Authors
I have already no comments